# Antibiotic and Copper Sensitivity in *Erwinia amylovora* Isolates from Northern Saudi Arabia, and the Induction of Fire Blight Suppression by Salicylic Acid

**DOI:** 10.3390/plants14203192

**Published:** 2025-10-17

**Authors:** Ali A. Al Masrahi, Abdurrehman M. Rafique, Abdullah F. Al Hashel, Mohammed A. Al Saleh, Yasser E. Ibrahim

**Affiliations:** Department of Plant Protection, College of Food and Agricultural Sciences, King Saud University, P.O. Box 2460, Riyadh 11451, Saudi Arabia; aalmasrahi@ksu.edu.sa (A.A.A.M.); arafiquf@ksu.edu.sa (A.M.R.); aalhashel@ksu.edu.sa (A.F.A.H.); malsaleh@ksu.edu.sa (M.A.A.S.)

**Keywords:** fire blight, streptomycin resistance, copper sulfate, salicylic acid, defense enzymes

## Abstract

Fire blight, caused by *Erwinia amylovora*, is a severe disease impacting pome fruit production worldwide, including in Saudi Arabia. This study evaluated antibiotic sensitivity and the potential of chemical and elicitor treatments to suppress *E. amylovora* isolates collected from various regions in Saudi Arabia. In the in vitro assays, at low antibiotic levels (10 µg/mL streptomycin and 25 µg/mL oxytetracycline), all Saudi Arabian strains exhibited minimal inhibition (zones ≤ 14 mm). Two isolates displayed partial tolerance at an intermediate oxytetracycline concentration (50 µg/mL). True sensitivity (zones > 18 mm) was mainly observed at the highest tested oxytetracycline dose (100 µg/mL). Regarding copper sulfate, all isolates showed no inhibition between 0.02 and 0.08 mM, while all isolates exhibited intermediate susceptibility at 0.16 mM. The second experimental phase examined in planta effects of streptomycin, salicylic acid (SA), and their combination on disease development in artificially inoculated apple (*Malus domestica*) shoots under greenhouse conditions. Both streptomycin and SA significantly reduced fire blight incidence (by 75%) and symptom severity, while the combined treatment yielded the greatest reduction in shoot necrosis and bacterial load. This is the first report demonstrating that SA, particularly when used in combination with streptomycin, can effectively suppress fire blight in Saudi Arabia. These results stress the importance of integrating resistance inducers into fire blight management strategies to counter the rise in antimicrobial resistance.

## 1. Introduction

Fire blight, caused by the Gram-negative bacterium *Erwinia amylovora* [1,2], is one of the most economically damaging diseases affecting pome fruit crops, especially apples (*Malus domestica* (Suckow) Borkh) and pears (*Pyrus communis* L.). It is classified as a quarantine pathogen in many countries, including those in the Middle East. The pathogen’s rapid spread and ability to move long distances via infected plant material pose a significant biosecurity risk [3,4]. The pathogen survives the winter in cankers on infected trees. It reappears in spring when temperatures exceed 18 °C, triggering new infections mainly through wounds or natural openings in blossoms, shoots, and leaves [5,6]. Spread is aided by rain, wind, and insect vectors [5]. In Saudi Arabia, fire blight was first confirmed in 2024, with subsequent outbreaks reported in several pome fruit-growing regions [7].

Fire blight causes significant economic losses, not only through the direct damage it inflicts on fruit trees but also due to the harsh measures often necessary to control it, such as removing entire orchards. One notable example is a severe outbreak in southwest Michigan, where 220,000 trees were lost, resulting in losses of $42 million [8]. Similarly, a significant outbreak in Switzerland in 2007 led to estimated losses of $27.5 million [9]. In Italy’s Emilia Romagna region, which accounts for 65% of the country’s pear production, over one million trees were destroyed between 1994 and 2004 to prevent the spread of the disease [10].

Historically, chemical control has been crucial in managing fire blight, primarily through antibiotics such as streptomycin and oxytetracycline. While streptomycin remains the most widely used antibiotic in the United States due to its bactericidal properties and cost-effectiveness [11,12], its effectiveness has been compromised by the emergence of resistant *E. amylovora* strains. Resistance was first reported in California in the 1970s and has since been identified in key fruit-producing regions worldwide, including New York, Oregon, Washington, Michigan, Egypt, New Zealand, Lebanon, and Mexico [11,12,13,14,15,16,17,18,19].

Oxytetracycline, although useful in semi-arid regions like the western U.S., is less effective in humid climates due to its bacteriostatic mode of action [9,20]. The Environmental Protection Agency registered the aminoglycoside antibiotic kasugamycin in 2015 (EPA registration number 66330-404). While it can offer a similar level of control, it lacks the curative potential and cost-effectiveness of streptomycin [17,21].

Copper-based compounds, such as copper sulfate, provide an alternative approach but are constrained by their phytotoxic effects on young fruits and foliage. Their use is typically limited to early-season sprays before flowering [10,22]. To date, streptomycin, oxytetracycline, and copper compounds are frequently employed in Saudi Arabia to manage fire blight disease. The increasing rate of antibiotic resistance, along with environmental concerns about heavy metal accumulation, underscores the need to explore sustainable methods of disease control.

Managing fire blight usually involves a mix of cultural, biological, and chemical methods [23]. However, the repeated use of antibiotics and copper-based treatments can lead to the development of resistance in *E. amylovora*, thereby reducing their effectiveness over time [24]. Consequently, biological control approaches are gaining popularity due to their potential to reduce reliance on chemicals [25,26]. Moreover, breeding and selecting fire blight-resistant cultivars is seen as one of the most sustainable and efficient strategies. Varieties such as Potomac, Old Home, and NJA2R59T69 have shown significant resistance to the disease [27].

One promising approach involves activating systemic acquired resistance (SAR) through compounds like Salicylic acid (SA) [28,29,30,31,32,33]. SA has been shown to initiate a broad range of defense mechanisms, including the accumulation of pathogenesis-related proteins, hydrogen peroxide, and defense-related enzymes such as peroxidase and polyphenol oxidase [29,30,31,32]. Previous research has demonstrated the potential of SA and its analoge (e.g., acibenzolar-S-methyl) to reduce disease severity and activate a strong defense-related response across various host–pathogen systems, including fire blight [34,35]. These treatments prime the host to respond more swiftly and effectively upon pathogen attack, a phenomenon known as “priming”.

Streptomycin and copper compounds remain central to fire blight management, yet widespread resistance has emerged due to repeated use [36]. Saudi Arabia-specific resistance data are limited. Additionally, induced resistance using molecules like SA or its analogs has shown promise elsewhere [34]; however, this approach has not yet been investigated in Saudi Arabia. This study aims to address this gap by: (1) characterizing resistance profiles of local isolates, and (2) assessing SA as a defense inducer against fire blight under greenhouse conditions.

## 2. Results

### 2.1. Antimicrobial Sensitivity of Erwinia amylovora Isolates from Northern Saudi Arabia

A total of 26 *E. amylovora* isolates from three regions of Saudi Arabia (Tabuk, Al-Jouf, and Hail) were evaluated for their sensitivity to streptomycin, oxytetracycline, and copper sulfate (Table 1). The growth of the Saudi Arabian strains was not inhibited by streptomycin and oxytetracycline at low concentrations (10 µg/mL and 25 µg/mL, respectively), as indicated by zones of inhibition ≤ 14 mm. At intermediate concentrations of oxytetracycline (50 µg/mL), partial tolerance was recorded in a limited number of isolates. In contrast, sensitivity emerged in most isolates only at the highest tested concentration (100 µg/mL), as indicated by zone diameters greater than 18 mm.

For copper sulfate, all isolates showed no inhibition between 0.02 and 0.08 mM, while all isolates exhibited intermediate susceptibility (+) at 0.16 mM. At the highest concentration of 1.10 mM, all isolates demonstrated complete sensitivity (–). Figure 1 illustrates the inhibition zones formed by representative isolates under varying antibiotic concentrations, highlighting the consistent resistance to streptomycin and oxytetracycline at lower dosages. Figure 2 compares the sensitivity to copper sulfate across all tested isolates, visually confirming the predominance of sensitivity and the limited incidence of tolerance.

### 2.2. Effect of Treatments on Shoot Infection by Erwinia amylovora

The effectiveness of different treatments in reducing shoot infection was assessed by quantifying the percentage of infected shoots under each condition. As illustrated in Figure 3, the combination of streptomycin and SA resulted in the lowest infection rate (~30%), significantly outperforming all other treatments (*p* < 0.05). Treatments with either streptomycin or SA alone reduced shoot infection to a similar extent (~40%), both significantly lower than the untreated infected control group, which exhibited the highest infection rate (~65%). Statistical groupings indicated by different letters above the bars confirm significant differences among treatment effects. The combination treatment (streptomycin + SA) formed a distinct group (“c”), reflecting its superior efficacy in suppressing infection, while streptomycin and salicylic acid alone belonged to group “b”. The infected untreated control was categorized as group “a”, underscoring its significantly higher infection level compared to all treatments. These results demonstrate a synergistic effect when streptomycin is combined with salicylic acid, enhancing disease control compared to individual treatments.

### 2.3. Effect of Treatments on Canker Development

Canker extension varied significantly among the different treatments (Figure 4). The infected untreated control exhibited the highest canker extension, reaching approximately 75%, and was significantly different from all other treatments (*p* < 0.05). Application of streptomycin alone reduced canker extension by about 66%, while the application of SA alone further reduced it to 53%. The greatest suppression of disease symptoms was observed with the combined treatment of streptomycin and SA, which limited canker development to approximately 39%. Statistical grouping (indicated by different letters above the bars) confirmed that the combination treatment (group d) was significantly more effective than either treatment alone (b and c), and all treatments differed significantly from the control (a). These results highlight the potential of integrated treatment strategies to reduce fire blight severity more effectively than single-agent applications.

### 2.4. Salicylic Acid Induces Defense-Related Enzyme Activities in Apple

To evaluate the role of SA in apple defense activation, the activities of peroxidase (POD), polyphenol oxidase (PPO), and hydrogen peroxide (H_2_O_2_) were quantified in healthy, infected, and treated plants.

#### 2.4.1. Peroxidase Activity

As shown in Figure 5A, peroxidase activity increased significantly in response to SA treatment, particularly when combined with infection. The highest enzyme activity was recorded in the SA + infected treatment (ΔA430 ≈ 0.25 m·min^−1^·mg^−1^), labeled statistically as group ‘a’. This was followed by SA alone (≈0.17), group ‘b’. Both infected and healthy controls exhibited significantly lower POD activity (~0.07 and ~0.05, respectively), grouped as ‘c’. This trend indicated that SA induces POD activity and that the pathogen challenge enhances this response synergistically.

#### 2.4.2. Polyphenol Oxidase Activity

A similar trend was observed for polyphenol oxidase (Figure 5B). The SA + infected treatment again produced the highest PPO activity (ΔA490 ≈ 0.105), significantly greater than all other groups (group ‘a’). SA alone increased PPO activity to approximately 0.075 (group ‘b’), while both infected and healthy plants exhibited minimal activity (~0.025 and ~0.015, respectively), grouped as ‘c’. These results suggest that PPO is strongly upregulated by SA, particularly in the presence of infection.

#### 2.4.3. Hydrogen Peroxide Accumulation

Hydrogen peroxide accumulation, a hallmark of oxidative burst during plant defense, also followed this pattern (Figure 5C). The combined SA + infected group showed the highest H_2_O_2_ levels (ΔA390 ≈ 0.09 µmol/g), labeled group ‘a’. SA alone resulted in moderate H_2_O_2_ production (~0.06), statistically distinct (group ‘b’). Both healthy and infected, untreated plants exhibited similarly low levels (~0.03), grouped as ‘c’. These findings confirm that SA primes apple plants for defense through oxidative stress mechanisms.

## 3. Discussion

The present study provides the first integrated evaluation of antibiotic sensitivity and elicitor-mediated disease suppression in *E. amylovora* isolates from Saudi Arabia. The consistent alignment between in vitro inhibition patterns and in planta disease reduction supports a functional relationship between antibiotic sensitivity and pathogen suppression under greenhouse conditions. Notably, this study also highlights a concerning trend of reduced sensitivity among *E. amylovora* isolates to commonly used antibiotics, particularly streptomycin and oxytetracycline. Growth of local strains was not inhibited by streptomycin and oxytetracycline at low concentrations (10 µg mL^−1^ and 25 µg mL^−1^, respectively). At 50 µg mL^−1^ oxytetracycline, two isolates exhibited partial tolerance, whereas marked sensitivity was observed in most isolates only at the highest tested level (100 µg mL^−1^). These findings parallel global reports that extensive and repeated antibiotic use has promoted resistance development in *E. amylovora* populations [11,12,13,14,15,16,17,18,19].

Streptomycin resistance is especially critical since it has long been central to fire blight management. Resistance mechanisms typically involve acquisition of the *strA–strB* genes, which can be transferred horizontally among bacterial populations [37,38]. The partial resistance detected in Saudi Arabia isolates suggests both misuse of antibiotics and possible genetic exchange [20]. Comparable findings in Europe and North America show that resistance emergence has reduced long-term streptomycin efficacy [20,39]. Similarly, oxytetracycline resistance—though less common—has been linked to ribosomal protection and efflux pump activity [36]. These patterns emphasize the need for continuous monitoring and rotation of bactericides to delay resistance spread.

Copper sensitivity assays showed that all Saudi Arabia isolates tolerated 0.16 mM CuSO_4_. Copper has long served as a broad-spectrum bactericide, but *E. amylovora* can persist under copper stress by entering a viable-but-nonculturable state [40] or by activating tolerance mechanisms such as the *copA* efflux pump [41]. Although oxytetracycline and copper were not used in subsequent greenhouse trials, their inclusion provided valuable baseline data on antimicrobial variability in Saudi Arabia isolates, information crucial for integrated fire blight management.

Importantly, the sensitivity screening provided the rationale for choosing streptomycin in the second part of this study, where we examined its combined effect with SA. Streptomycin remained the most effective compound across isolates, and despite emerging resistance, it is still the most widely applied antibiotic against fire blight. The observed synergistic effect of combining streptomycin with SA under greenhouse conditions resulted in the lowest infection rates (~30%). This suggests that integrating host resistance inducers such as SA with conventional bactericides could lower reliance on antibiotics, delay resistance development, and provide a more sustainable disease management strategy. SA can directly inhibit bacterial growth by affecting virulence factors, motility, and biofilm formation, while also promoting host defense responses [42]. Although some pathogens degrade SA to overcome this defense [43], its dual role as antimicrobial and inducer of systemic acquired resistance makes it a strong candidate for integration with antibiotics [28,34].

Our results showed significant upregulation of defense-related markers, including POD, PPO, and H_2_O_2_, particularly in SA-treated plants. For instance, POD activity nearly tripled compared with controls. These findings align with earlier studies linking SAR to oxidative bursts and enzyme accumulation, which strengthen cell walls and hinder pathogen invasion [33,44]. The observed synergy therefore likely results from both direct bacterial suppression and enhanced plant immunity.

While promising, the combined use of SA and antibiotics also raises environmental considerations. Streptomycin and oxytetracycline residues may disrupt beneficial microbial communities in soil, phyllosphere, and pollinators [36,45]. Repeated use could also promote resistance in non-target bacteria [3]. Although SA may help reduce antibiotic doses, exogenous SA itself can alter microbial community composition, favoring tolerant species and potentially affecting soil nutrient cycling or beneficial symbioses [33]. Thus, field validation should include monitoring of non-target effects and ecosystem impacts in addition to efficacy.

## 4. Materials and Methods

### 4.1. Bacterial Isolates Collection and Identification

A total of twenty-six *Erwinia amylovora* isolates were obtained from symptomatic pear trees in Al-Jouf, Hail, and Tabuk regions during 2020–2022 [7]. Prior to these years, these isolates had been characterized and identified through the standard morphological, biochemical, and molecular techniques which included the *pEA29* plasmid PCR amplification, and *16S rRNA* and *rpoB* gene sequencing. Sequence data for these isolates are available in GenBank under accession numbers OR717505, OR743536–OR743560, and PP465516–PP465541 [7]. The isolates are part of the King Saud University, Plant Pathology Culture Collection (KSU-PCC) in Riyadh, Saudi Arabia. Prior to use in experiments, the cultures were routinely subcultured on sucrose nutrient agar (SNA) to ensure sufficient growth.

### 4.2. Antibiotic Sensitivity Testing

Streptomycin and oxytetracycline sensitivity were tested using the disk diffusion method. Pure cultures of the 26 isolates were routinely grown at 26 °C on SNA plates [46]. Single colonies from 48-h cultures were suspended in sterile distilled water (SDW) and adjusted to 108 colony-forming units per milliliter (A640 = 0.10 − 0.15). One hundred microliters of bacterial suspension were spread on the surface of the SNA medium. Autoclaved filter paper disks were impregnated with 0, 10, 100, 500, and 1000 µg/mL of streptomycin sulfate or oxytetracycline sulfate at 0, 10, 25, 50, or 100 µg/mL, and then placed on SNA containing the bacterial lawns [47,48]. Plates were incubated at 26 °C for 48 h. Each isolate was tested twice, with four replicates per treatment.

### 4.3. Copper Sensitivity Testing

Copper sulfate tolerance was assessed on casitone-yeast extract (CYE) medium, which consisted of 1.7 g casitone, 0.35 g yeast extract, 2 g glucose, and 15 g agar in 1000 mL of deionised water, as described by [48]. CYE agar was supplemented with CuSO_4_ at concentrations of 0.02, 0.04, 0.08, 0.16, and 1.10 mM, as previously reported by [48]. These concentrations were selected based on established research indicating their relevance for bacterial growth inhibition and field application rates [48]. Isolates were spotted onto the plates and incubated at 26 °C. Bacterial growth was recorded after 72 h. Control plates without copper served as the baseline. All tests were conducted in four replicates.

### 4.4. Greenhouse Evaluation of SA and Streptomycin

#### 4.4.1. Apple Seedlings

Greenhouse experiments were conducted on the apple (*Malus domestica*) cv. Gala and 2-year-old Gala scions grafted onto M9 rootstock seedlings (8–10 leaves). Plants were grown in 3.8 L pots using a 1:2 mixture of soil and commercial peat, and were supplemented every 10 days with a 10-52-10 fertilizer at a rate of 2 g/L (N-P-K).

#### 4.4.2. Inoculum Preparation

A virulent strain, Ea. 06 of *E. amylovora* [7], was used for inoculation. Bacteria were cultured for 24 h at 24 °C on King’s medium B [49]. Bacterial suspensions were prepared in SDW to a final concentration of 1 × 10^9^ CFU/mL, as determined using a spectrophotometer at a wavelength corresponding to an optical density (OD600) of 0.62.

#### 4.4.3. Greenhouse Experiments

Apple seedlings were sprayed twice with SA (10 mM), at 7 and 14 days before inoculation. The inoculation was performed by cutting the tips (about 1.5 cm from the tip) below the first undeveloped leaf with scissors previously immersed for 20 s in a water suspension of *E. amylovora* strain Ea. 06 containing 1 × 10^9^ CFU mL^−1^ [50]. For the control seedlings, sterile distilled water was used instead of bacterial suspension. The exact number of plants was sprayed with the standard streptomycin sulfate concentration of 100 mg a.i./L immediately after wounding and allowed to dry for 4 h before inoculation. Infected shoots and canker length were recorded three weeks post-inoculation. Ten shoots were inoculated per treatment, and the experiment was repeated twice. Trials were conducted in a randomized complete block design with three replications.

### 4.5. Defense Enzymes and Hydrogen Peroxide Assays

Plant samples for analysis were collected 7 days post-inoculation from shoots approximately 5 cm in length, from 2 cm above and below the inoculation site. Approximately 500 mg of shoot tissue was homogenized in 2 mL of ice-cold extraction buffer (20 mM Tris-HCl buffer, pH 7.8, with 10% glycerol, 10% Triton X-100, 5% PEG 4000, and 1% NaCl). The crude extract was then centrifuged at 1500g for 15 min at 4 °C, and the supernatants were used for further analysis [51].

#### 4.5.1. Peroxidase (POX) Assay

POX activity was measured at room temperature using the method described by Klessig et al. (2001) [52]. The reaction mixture consisted of 50 mM pyrogallol in 50 mM phosphate buffer (pH 6), enzyme extract, and 3% hydrogen peroxide. The enzyme activity was measured as a change in absorbance at 430 nm over one minute (ΔA430/min/gram of fresh tissues).

#### 4.5.2. Polyphenol Oxidase (PPO) Assay

PPO activity was determined at 25 °C using catechol as a substrate [53]. The reaction mixture comprised 50 mM catechol in 50 mM phosphate buffer (pH 6.5) with the enzyme extract. PPO activity was assessed by measuring changes in absorbance at 430 nm over one minute (ΔA430/min/gram fresh tissue).

#### 4.5.3. Hydrogen Peroxide (H_2_O_2_) Assay

For the H_2_O_2_ assay, 0.5 g of the samples was homogenized in 0.1% Trichloroacetic acid and centrifuged at 12,000 rpm for 5 min at 4 °C. The supernatant (0.3 mL) was mixed with 1.7 mL of potassium phosphate buffer (pH 7.0) and 1 mL of 1 M potassium iodide solution, and then incubated for 5 min. The oxidation product was measured at A390, and concentration was calculated using a standard curve with a known concentration of H_2_O_2_, expressed as μmol/g [54].

### 4.6. Statistical Analysis

Percentages of infection were transformed using arcsin for statistical analysis. Before each analysis, the homogeneity of variances was checked with Bartlett’s test at a significance level of *p* = 0.05. A one-way analysis of variance (ANOVA) was applied to detect significant differences among treatments, followed by Duncan’s multiple range test to separate.

## 5. Conclusions

This study demonstrates the emergence of antibiotic resistance in Saudi Arabia *E. amylovora* populations and supports integrating chemical sensitivity data with host-induced resistance strategies. The two parts of this work, antibiotic sensitivity screening and greenhouse evaluation of SA-streptomycin combinations, are complementary: the former identified resistance patterns, while the latter showed how integrating antibiotics with resistance inducers can improve control, even against partially resistant isolates. Field-level trials are now essential to confirm the durability of these strategies and assess environmental safety. Practically, integrating SA with streptomycin could reduce chemical input, improve sustainability, and fit within Integrated Pest Management frameworks, particularly in Saudi Arabian orchards where fire blight remains a serious challenge.

## Figures and Tables

**Figure 1 plants-14-03192-f001:**
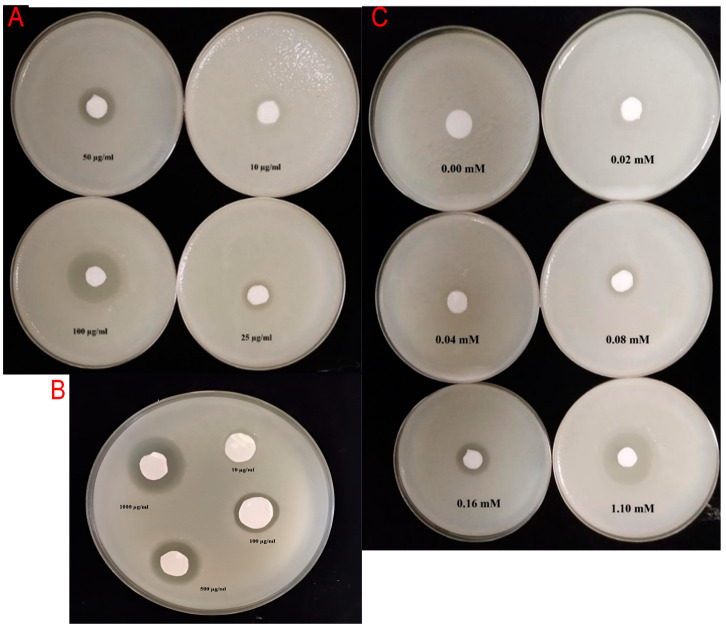
In vitro inhibition of *Erwinia amylovora* strain Ea06 by different concentrations of (**A**) oxytetracycline, (**B**) streptomycin sulfate, and (**C**) copper sulfate using the agar diffusion assay. Sterile paper disks were impregnated with the indicated concentrations of each compound and placed on nutrient agar plates previously inoculated with a bacterial suspension (≈10^8^ CFU mL^−1^). Plates were incubated at 28 °C for 48 h, and zones of growth inhibition were recorded as a measure of antibacterial activity. Each treatment was performed in triplicate, and representative plates are shown.

**Figure 2 plants-14-03192-f002:**
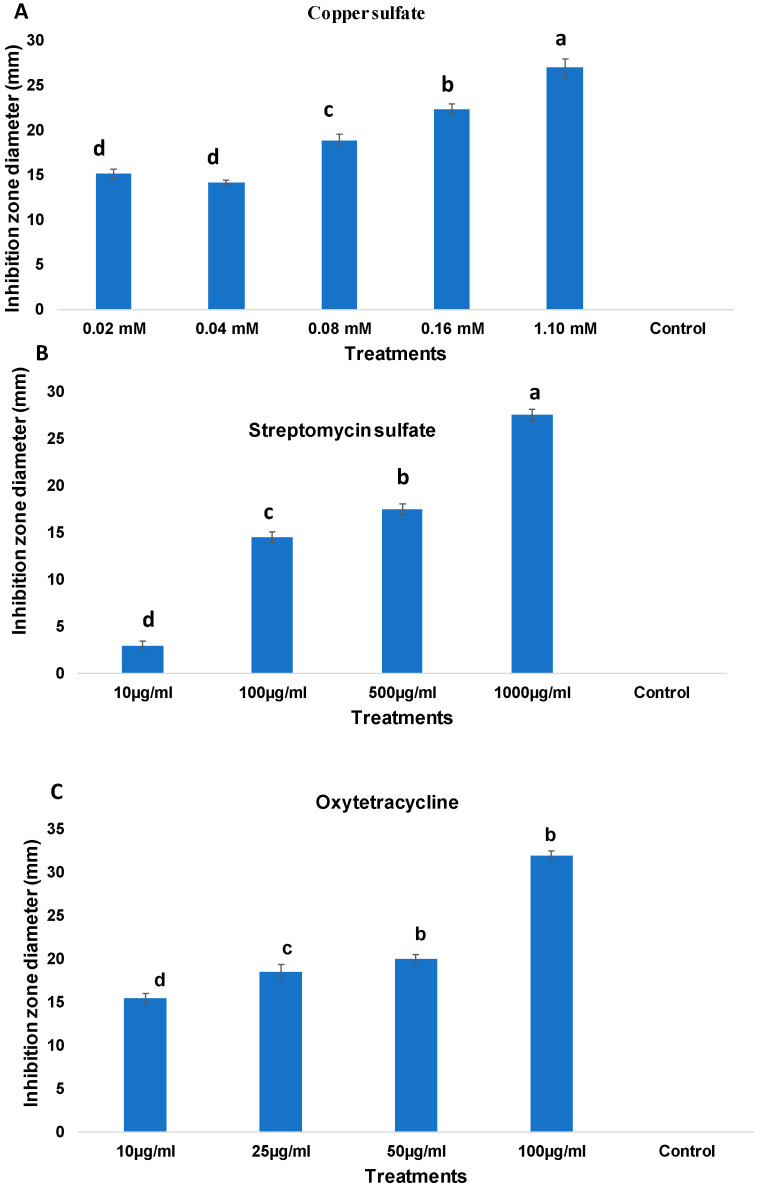
Growth inhibition of *Erwinia amylovora* strain Ea06 (1 × 10^9^ CFU/mL) by different concentrations of (**A**) copper sulfate, (**B**) streptomycin sulfate, and (**C**) oxytetracycline. Inhibition zones (mm) were measured after 48 h of incubation at 28 °C following the agar disk diffusion assay. Bars represent the mean ± standard error of three independent replicates. Different letters above the bars indicate significant differences among treatments according to Duncan’s multiple range least significant difference (LSD) test at *p* ≤ 0.05.

**Figure 3 plants-14-03192-f003:**
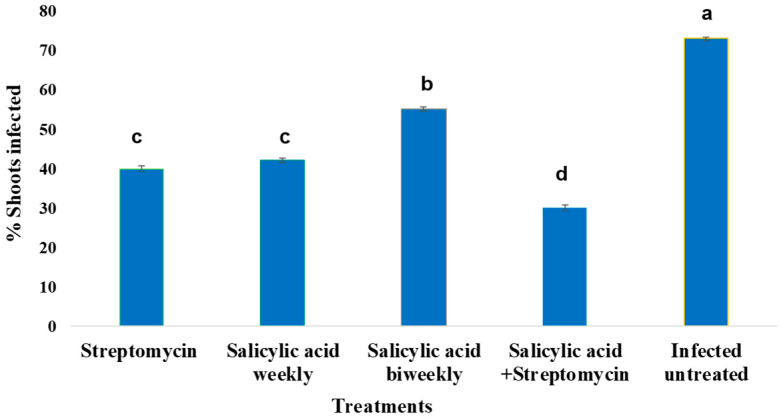
Effect of different treatments on the incidence of fire blight in ‘Gala’ apple (*Malus domestica*) shoots inoculated with *Erwinia amylovora* strain Ea06 (1 × 10^9^ CFU/mL). Bars represent the mean percentage of infected shoots per treatment, assessed two weeks after inoculation. Treatments included streptomycin, salicylic acid applied weekly or biweekly, their combination (salicylic acid + streptomycin), and an untreated infected control. Values are means ± standard error of three replicates. Columns labeled with the same letter are not significantly different according to Duncan’s multiple range least significant difference (LSD) test at *p* ≤ 0.05.

**Figure 4 plants-14-03192-f004:**
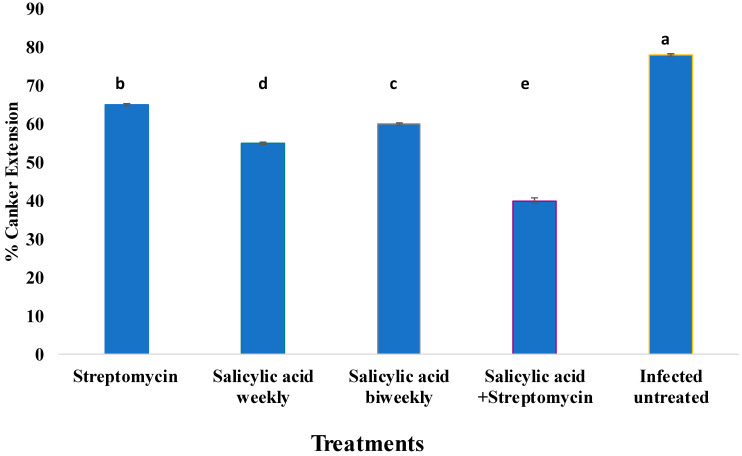
Effect of different treatments on fire blight canker extension in ‘Gala’ apple (*Malus domestica*) shoots inoculated with *Erwinia amylovora* strain Ea06 (1 × 10^9^ CFU/mL). Bars represent the mean percentage of canker extension measured two weeks post-inoculation. Treatments included streptomycin, salicylic acid applied weekly or biweekly, a combination of salicylic acid and streptomycin, and an infected untreated control. The combination treatment showed the greatest reduction in canker development compared with all other treatments. Bars with different letters indicate statistically significant differences according to Duncan’s multiple range least significant difference (LSD) test at *p* ≤ 0.05.

**Figure 5 plants-14-03192-f005:**
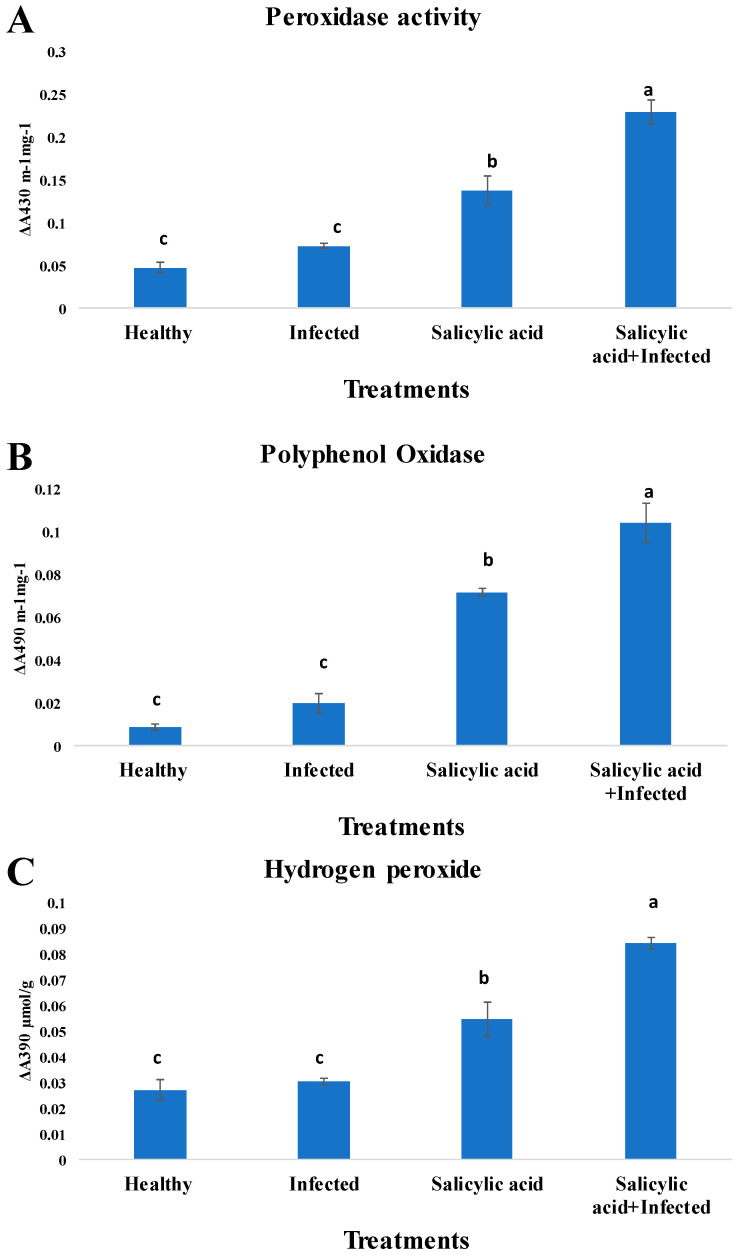
Activities of (**A**) peroxidase, (**B**) polyphenol oxidase, and (**C**) hydrogen peroxide in one-year-old ‘Gala’ apple (*Malus domestica*) seedlings infected with *Erwinia amylovora* strain Ea06 (1 × 10^9^ CFU/mL) and treated with salicylic acid under greenhouse conditions. Bars represent mean enzyme activity values, measured as changes in absorbance per unit time. Treatments included healthy control, infected control, salicylic acid alone, and the combination of salicylic acid with infection. Bars sharing the same letter within each panel are not significantly different (*p* > 0.05) according to Duncan’s multiple range least significant difference (LSD) test at *p* ≤ 0.05.

**Table 1 plants-14-03192-t001:** Sensitivity of 26 Saudi Arabian isolates of *Erwinia amylovora* against different concentrations of streptomycin, oxytetracycline, and copper sulfate.

Isolates	Streptomycin (µg/mL)					Oxytetracycline (µg/mL)				Copper Sulfate (mM)					
	0.0	10	100	500	1000	10	25	50	100	0.00	0.02	0.04	0.08	0.16	1.10
Ea01_Tabuk	-	+	-	-	-	-	-	-	-	-	+	+	+	(+)	-
Ea02_Tabuk	-	+	-	-	-	-	-	-	-	-	+	+	+	(+)	-
Ea03_Tabuk	-	+	-	-	-	-	-	-	-	-	+	+	+	(+)	-
Ea04_Tabuk	-	+	-	-	-	-	-	-	-	-	+	+	+	(+)	-
Ea05_Tabuk	-	+	-	-	-	-	-	-	-	-	+	+	+	(+)	-
Ea06_Tabuk	-	+	(+)	(+)	-	-	-	-	-	-	+	+	+	(+)	-
Ea07_Tabuk	-	+	-	-	-	-	-	-	-	-	+	+	+	(+)	-
Ea08_Tabuk	-	+	-	-	-	-	-	-	-	-	+	+	+	(+)	-
Ea09_Tabuk	-	+	-	-	-	-	-	-	-	-	+	+	+	(+)	-
Ea10_Al-Jouf	-	+	-	-	-	-	-	-	-	-	+	+	+	(+)	-
Ea11_Al-Jouf	-	+	-	-	-	-	-	-	-	-	+	+	+	(+)	-
Ea12_Al-Jouf	-	+	-	-	-	-	-	-	-	-	+	+	+	(+)	-
Ea13_Al-Jouf	-	+	-	-	-	-	-	-	-	-	+	+	+	(+)	-
Ea14_Al-Jouf	-	+	-	-	-	-	(+)	(+)	-	-	+	+	+	(+)	-
Ea15_Al-Jouf	-	+	-	-	-	-	-	-	-	-	+	+	+	(+)	-
Ea16_Al-Jouf	-	+	-	-	-	-	-	-	-	-	+	+	+	(+)	-
Ea17_Hail	-	+	-	-	-	-	-	-	-	-	+	+	+	(+)	-
Ea18_Hail	-	+	-	-	-	-	-	-	-	-	+	+	+	(+)	-
Ea19_Hail	-	+	-	-	-	-	-	-	-	-	+	+	+	(+)	-
Ea20_Hail	-	+	-	-	-	-	-	-	-	-	+	+	+	(+)	-
Ea21_Hail	-	+	-	-	-	-	-	-	-	-	+	+	+	(+)	-
Ea22_Hail	-	+	-	-	-	-	-	-	-	-	+	+	+	(+)	-
Ea23_Al-Jouf	-	+	-	-	-	-	-	-	-	-	+	+	+	(+)	-
Ea24_Al-Jouf	-	+	-	-	-	-	-	-	-	-	+	+	+	(+)	-
Ea25_Al-Jouf	-	+	-	-	-	-	-	-	-	-	+	+	+	(+)	-
Ea26_Al-Jouf	-	+	(+)	(+)	-	-	-	-	-	-	+	+	+	(+)	-

+ Resistant (zone diameter ≤ 12 mm for antibiotics and ≤20 mm for copper); (+) intermediate susceptible (zone diameter from 13–20 mm for antibiotics and from 20–25 mm for copper); - strong susceptible (no growth or ˃20 mm for antibiotics and ˃25 mm for copper) [35].

## Data Availability

The original contributions presented in this study are included in the article. Further inquiries can be directed to the corresponding author.

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
