# Peer review of "Antibiotic and Copper Sensitivity in Erwinia amylovora Isolates from Northern Saudi Arabia, and the Induction of Fire Blight Suppression by Salicylic Acid"

_plants, 2025, doi:10.3390/plants14203192_

Round 1
Reviewer 1 Report
Comments and Suggestions for Authors
- Why is the manuscript titled "Antibiotic and copper resistance..." when the authors state at the beginning of the Results section (lines 189-191) that the strains were tested for susceptibility to antibiotics and copper sulfate and demonstrated sensitivity to these toxicants (Table 1)? Why is "resistance" in the manuscript title, not "sensitivity"?
- Salicylic acid should be added to the keywords.
- Section 2.1 should be completely revised. The authors did not perform taxonomic identification of the isolates in this work. The bacterial collection in which the studied strains are deposited should be indicated here.
- Line 127: Why were these specific concentrations of copper sulfate used?
- Lines 138 and 147: Why was the inoculum prepared with strain E. 06, while the leaf treatment was performed with a suspension of strain Ea 06? Why was this strain used in the plant experiments and not the isolates described in Table 1? Ibrahim et al. (2024) stated that all strains were virulent. Why did the authors choose one strain?
- The quality of the figures should be improved.
- All figure captions should indicate which bacterial strain was used.
- Figure 5 should be presented as a diagram.
- The authors do not discuss the effect of salicylic acid on the bacterial strain they used and on plant growth. SA is an inhibitor of plant and bacterial growth.
- Overall, the manuscript describes two studies: i) a study of the sensitivity of 26 bacterial isolates to streptomycin, tetracycline, and copper sulfate; ii) a study of the phenomenon and enzymatic mechanism of infection reduction with the combined action of streptomycin and salicylic acid and infection with a single virulent strain. These two parts are poorly coordinated, and the authors do not connect them in any way. Why conduct sensitivity testing to tetracycline and copper if it was not used further? In my opinion, both parts should be described in two separate manuscripts. Alternatively, the authors should present the results of experiments with tetracycline and salicylic acid, as well as with copper sulfate and salicylic acid.
Author Response
Reviewr#1
Comment: Why is the manuscript titled "Antibiotic and copper resistance..." when the authors state at the beginning of the Results section (lines 189-191) that the strains were tested for susceptibility to antibiotics and copper sulfate and demonstrated sensitivity to these toxicants (Table 1)? Why is "resistance" in the manuscript title, not "sensitivity"?
Response: The word resistance changed to sensitivity in the title
Comment: Salicylic acid should be added to the keywords.
Response: Salicylic acid was added to the keywords
Comment: Section 2.1 should be completely revised. The authors did not perform taxonomic identification of the isolates in this work. The bacterial collection in which the studied strains are deposited should be indicated here.
Response: This section was improved based on the reviewer comment in lines 293-301
Comment: Line 127: Why were these specific concentrations of copper sulfate used?
Response: Corrected in lines 315-317
Comment: Lines 138 and 147: Why was the inoculum prepared with strain E. 06, while the leaf treatment was performed with a suspension of strain Ea 06? Why was this strain used in the plant experiments and not the isolates described in Table 1? Ibrahim et al. (2024) stated that all strains were virulent. Why did the authors choose one strain?
Response: The apparent discrepancy between “E. 06” and “Ea 06” is due to a typographical oversight. In the manuscript, all isolates are labeled with the prefix “E” in Table 1, but the full designation should have included “a” (i.e., Ea 06) and all fixed in Table 1 and text through the manuscript. The inoculum prepared and the leaf treatments were performed with the same strain, Ea 06. Although all isolates were confirmed to be virulent (Ibrahim et al., 2024), Ea 06 was selected for the plant experiments because it exhibited the highest virulence in pathogenicity tests.
Comment: The quality of the figures should be improved.
Response: All figures improved
Comment: All figure captions should indicate which bacterial strain was used.
Response: The bacterial strain was added in all figure’s captions
Comment: Figure 5 should be presented as a diagram.
Response: Fig. 5 was presented as a diagram
Comment: The authors do not discuss the effect of salicylic acid on the bacterial strain they used and on plant growth. SA is an inhibitor of plant and bacterial growth.
Response: Fixed in lines 271-273
Comment: Overall, the manuscript describes two studies: i) a study of the sensitivity of 26 bacterial isolates to streptomycin, tetracycline, and copper sulfate; ii) a study of the phenomenon and enzymatic mechanism of infection reduction with the combined action of streptomycin and salicylic acid and infection with a single virulent strain. These two parts are poorly coordinated, and the authors do not connect them in any way. Why conduct sensitivity testing to tetracycline and copper if it was not used further? In my opinion, both parts should be described in two separate manuscripts. Alternatively, the authors should present the results of experiments with tetracycline and salicylic acid, as well as with copper sulfate and salicylic acid.
Response:
Why conduct sensitivity testing to tetracycline and copper if it was not used further?
The sensitivity assays to streptomycin, tetracycline, and copper sulfate were performed to provide a complete baseline profile of the Saudi Arabian E. amylovora isolates. These antimicrobials are among the most widely used or historically tested agents in fire blight management (Loper, 1991; McManus et al., 2002; Stockwell et al. 2008). Presenting the full panel was essential to characterize the isolates and to highlight potential alternatives in case of streptomycin resistance.
Why streptomycin was chosen for combination with SA:
Although tetracycline and copper sulfate were tested in vitro, we focused subsequent combination experiments on streptomycin because:
Streptomycin remains the most widely applied antibiotic against fire blight in many regions, including Saudi Arabia, and resistance development is a key concern (Ibrahim et al., 2024).
Our preliminary screening showed that all isolates were moderately to highly sensitive to streptomycin, making it the most suitable candidate for evaluating the additive/synergistic effect of SA.
Copper and tetracycline have known limitations in planta (phytotoxicity for copper; instability and regulatory restrictions for tetracycline), which restricted their use in greenhouse assays.
Why the manuscript was kept unified:
We believe presenting both parts within one manuscript strengthens the study, as it links the antibiotic resistance profile of local isolates to a novel integrated treatment strategy. The sensitivity data provide essential context for why streptomycin was selected for further testing with SA, ensuring a logical progression from screening to application.
Alternatively, the authors should present the results of experiments with tetracycline and salicylic acid, as well as with copper sulfate and salicylic acid.
The dissection section was modified in lines 264-276.
Reviewer 2 Report
Comments and Suggestions for Authors
I have read this paper. The study presents a comprehensive examination of antibiotic and copper resistance in Erwinia amylovora isolate from northern Saudi Arabia, with evaluating the role of salicylic acid (SA) in suppressing fire blight. This study deal with an important agricultural issue, regarding the growing resistance among pathogens to conventional chemical treatments, which is critical to sustainable farming practices. Both lab and greenhouse experiments provides solid evidence for potential application of SA as a complementary treatment to traditional antibiotics in managing fire blight.
General comments
Explain in detail statistical analyses and the practical implications of the results in a broader agricultural context.
Field results will further enhance the applicability.
Quality of the figures need to improve.
Table one is very long adjust it properly.
Specific comments
Add more recent studies in the introduction section particularly related to the ongoing research in fire blight management in Saudi Arabia. (https://www.sciencedirect.com/science/article/pii/S2667064X2500185X; https://www.frontiersin.org/journals/sustainable-food-systems/articles/10.3389/fsufs.2025.1581883/full)
Why use ANOVA? There are multiple tests are available. Explain the choice for your data let the reader understand it.
Figures 1, 2….etc, improve the quality. I can’t read them.
What are the protentional environmental impacts of using SA in combination with antibiotics? Explain in discussion section.
Figures 1-4 should have more detail captions
Add study limitations and how local farmers can benefit from it?
Comments on the Quality of English Language
I suggest language editing
Author Response
Reviewr#2
I have read this paper. The study presents a comprehensive examination of antibiotic and copper resistance in Erwinia amylovora isolate from northern Saudi Arabia, with evaluating the role of salicylic acid (SA) in suppressing fire blight. This study deal with an important agricultural issue, regarding the growing resistance among pathogens to conventional chemical treatments, which is critical to sustainable farming practices. Both lab and greenhouse experiments provides solid evidence for potential application of SA as a complementary treatment to traditional antibiotics in managing fire blight.
General comments
Comment: Explain in detail statistical analyses and the practical implications of the results in a broader agricultural context.
Response: Done in lines 283-290
Field results will further enhance the applicability.
Response: Done in lines 286-290
Quality of the figures need to improve.
Response: Improved
Table one is very long adjust it properly.
Response: Improved
Specific comments
Add more recent studies in the introduction section particularly related to the ongoing research in fire blight management in Saudi Arabia. (https://www.sciencedirect.com/science/article/pii/S2667064X2500185X; https://www.frontiersin.org/journals/sustainable-food-systems/articles/10.3389/fsufs.2025.1581883/full)
Response: Done in an introduction section line 74.
Why use ANOVA? There are multiple tests are available. Explain the choice for your data let the reader understand it.
Response: Data from the sensitivity assays and pathogenicity experiments were analyzed using ANOVA because the experimental design included multiple treatments with more than two groups to compare simultaneously. ANOVA is a robust method to detect statistically significant differences in mean responses across treatments while controlling the type I error rate, which would be inflated if multiple pairwise t-tests were applied instead. When significant effects were detected, means were further separated using Tukey’s HSD test to account for multiple comparisons. The use of ANOVA in this study is consistent with previous plant pathology and bacteriology research where treatment effects across several concentrations or strains are compared (Gomez & Gomez, 1984).
Figures 1, 2…. etc, improve the quality. I can’t read them.
Response: Improved
What are the protentional environmental impacts of using SA in combination with antibiotics? Explain in discussion section.
Response: Done in lines 283-290
Figures 1-4 should have more detail captions
Response: Fixed and more detail were added
Add study limitations and how local farmers can benefit from it?
Response: Fixed in lines 283-290
Reviewer 3 Report
Comments and Suggestions for Authors
The manuscript presents the results of the resistance profile of multiple Erwinia amylovora strains, the causal agent of fire blight, to antibiotics and copper. Furthermore, the Authors investigated the effectiveness of antibiotics combined with salicylic acid against infection symptoms, as well as the influence of treatments on plant defence mechanisms. I find the study interesting and scientifically valuable. Both the Results and Discussion sections are rather concise, which I generally consider advantageous. However, I have several issues that should be addressed by the Authors:
-
Materials and Methods 2.4.2 – What do you mean by “as determined using a spectrophotometer at a wavelength corresponding to an optical density (OD600) of 0.62”? Isn’t 600 nm wavelength that you used?
-
Figure 2 – The quality is too low; the text is not legible. A more detailed description in the figure legend would also be helpful (e.g., what concentrations were used).
-
Please avoid repeating results in the Discussion (e.g., line 292).
-
Line 307 – “Tolerance to what?” (It seems the word copper is missing in this sentence.)
-
Please provide one strong concluding statement regarding the results obtained for copper compounds.
-
Lines 324–325 – In my opinion, this paragraph is underdiscussed and should be elaborated further. The described mechanisms are very interesting, and more detail would add significant value.
Author Response
Reviewr#3
The manuscript presents the results of the resistance profile of multiple Erwinia amylovora strains, the causal agent of fire blight, to antibiotics and copper. Furthermore, the Authors investigated the effectiveness of antibiotics combined with salicylic acid against infection symptoms, as well as the influence of treatments on plant defence mechanisms. I find the study interesting and scientifically valuable. Both the Results and Discussion sections are rather concise, which I generally consider advantageous. However, I have several issues that should be addressed by the Authors:
Comment: Materials and Methods 2.4.2 – What do you mean by “as determined using a spectrophotometer at a wavelength corresponding to an optical density (OD600) of 0.62”? Isn’t 600 nm wavelength that you used?
Response: We thank the reviewer for pointing out this ambiguity. Indeed, we measured bacterial cell density at a wavelength of 600 nm, which corresponds to the commonly used optical density parameter OD600. To clarify, we will revise the sentence in the Materials and Methods section as follows:
“Bacterial cultures were adjusted to a final concentration corresponding to an optical density of 0.62 at 600 nm (OD600), measured using a spectrophotometer.” Lines 329-331.
Comment: Figure 2 – The quality is too low; the text is not legible. A more detailed description in the figure legend would also be helpful (e.g., what concentrations were used).
Response: The quality of all figures was improved. More detailed description in figure legend were added.
Comment: Please avoid repeating results in the Discussion (e.g., line 292).
Response: Fixed
Comment: Line 307 – “Tolerance to what?” (It seems the word copper is missing in this sentence.)
Response: Fixed in this section
Comment: Please provide one strong concluding statement regarding the results obtained for copper compounds.
Response: The discussion section was adjusted and highlighted with red color
Lines 324–325 – In my opinion, this paragraph is underdiscussed and should be elaborated further. The described mechanisms are very interesting, and more detail would add significant value.
Response: The discussion section was adjusted and highlighted with red color
Round 2
Reviewer 1 Report
Comments and Suggestions for Authors
The authors responded to all my comments. The two parts of the paper still seem poorly connected to me. Perhaps the abstract should clarify that the experiments in the first part describe the results of bacterial growth on agar nutrient medium, while the experiments in the second part describe the results on plants.
Minor comments:
1. Plant names should be written with their full scientific names when first mentioned - Malus domestica (Suckow) Borkh.; Pyrus communis L., etc.
2. Why do the authors show in Table 1 that without the addition of streptomycin and copper to the medium, zone diameters of ≤ 12 mm for antibiotics and ≤ 20 mm for copper were observed? These are controls. There should be no inhibition zones. These data are not provided for tetracycline.
3. Throughout the manuscript, the authors repeatedly abbreviate "Saudi Arabia" to "Saudi." I find this abbreviation inappropriate when indicating the location of bacterial isolation or plant growth.
4. The authors use two terms for the same concept: absorbance and optical density, denoting them as A and OD. Use a single notation.
Author Response
Response to Reviewer Comments
We thank the reviewer for the thoughtful and constructive feedback. All comments have been carefully considered, and the manuscript has been revised accordingly. Below, we provide detailed point-by-point responses.
General Comment
The two parts of the paper still seem poorly connected to me. Perhaps the abstract should clarify that the experiments in the first part describe the results of bacterial growth on agar nutrient medium, while the experiments in the second part describe the results on plants.
Response:
We appreciate this important observation. The Abstract has been revised to clearly distinguish between the two experimental components. It now explicitly states that the first section presents in vitro bacterial growth and antibiotic sensitivity assays on nutrient agar, while the second section presents in planta pathogenicity and control experiments. Also, the First paragraph in the discussion was added in lines 241-245.
Minor Comment 1
Plant names should be written with their full scientific names when first mentioned – Malus domestica (Suckow) Borkh.; Pyrus communis L., etc.
Response:
We thank the reviewer for pointing this out. All plant species mentioned in the manuscript are now written with their full binomial scientific names and highlighted with red color in lines 34-35.
Minor Comment 2
Why do the authors show in Table 1 that without the addition of streptomycin and copper to the medium, zone diameters of ≤ 12 mm for antibiotics and ≤ 20 mm for copper were observed? These are controls. There should be no inhibition zones. These data are not provided for tetracycline.
Response:
We appreciate this observation and have rechecked our Table and corrected
Minor Comment 3
Throughout the manuscript, the authors repeatedly abbreviate “Saudi Arabia” to “Saudi.” I find this abbreviation inappropriate when indicating the location of bacterial isolation or plant growth.
Response:
We thank the reviewer for noting this stylistic issue. All instances of the abbreviation “Saudi” have been replaced with the full term “Saudi Arabia” to maintain formal geographic naming consistency.
Minor Comment 4
The authors use two terms for the same concept: absorbance and optical density, denoting them as A and OD. Use a single notation.
Response:
|
Absorbance (A₄₃₀) |
For biochemical enzyme assays (e.g., PPO, POD, CAT) |
Measures a chemical reaction — the formation of colored quinones absorbing light at a specific wavelength (430 nm). |
|
Optical Density (OD₆₀₀) |
For cell/turbidity measurements |
Measures light scattering by bacterial cells at 600 nm; not a chemical absorbance. |
Reviewer 2 Report
Comments and Suggestions for Authors
no more comments
Author Response
Dear Editor and Reviewers,
We sincerely thank you for thoughtful and constructive feedback on our manuscript. In accordance with your request, we have revised the manuscript comprehensively, addressing all comments and questions that were mentioned in the reviews. All modifications are clearly marked in red within the revised manuscript.
In addition to the changes made to the manuscript, we provide detailed point-by-point responses to all reviewer’s remarks in the accompanying tables. We very much appreciate the reviewers’ insights, which have significantly enhanced the quality and clarity of our work.
We submit this revised version for your consideration and remain hopeful that it now meets the standards for publication in your esteemed journal.
Thank you for your continued support and consideration.
Yours sincerely,